# Nanotechnology-Based Strategies to Combat Multidrug-Resistant *Candida auris* Infections

**DOI:** 10.3390/pathogens12081033

**Published:** 2023-08-13

**Authors:** Helal F. Hetta, Yasmin N. Ramadan, Israa M. S. Al-Kadmy, Noura H. Abd Ellah, Lama Shbibe, Basem Battah

**Affiliations:** 1Department of Medical Microbiology and Immunology, Faculty of Medicine, Assiut University, Assiut 71515, Egypt; helalhetta@aun.edu.eg; 2Department of Microbiology and Immunology, Faculty of Pharmacy, Assiut University, Assiut 71515, Egypt; yasmine_mohamed@pharm.aun.edu.eg; 3Branch of Biotechnology, Department of Biology, College of Science, Mustansiriyah University, Baghdad P.O. Box 10244, Iraq; israaalkadmy@gmail.com; 4Department of Pharmaceutics, Faculty of Pharmacy, Assiut University, Assiut 71515, Egypt; nora.1512@aun.edu.eg; 5Department of Pharmaceutics, Faculty of Pharmacy, Badr University in Assiut, Naser City, Assiut 2014101, Egypt; 6Faculty of Science, Damascus University, Damascus 97009, Syria; menaayoub1990@gmail.com; 7Department of Biochemistry and Microbiology, Faculty of Pharmacy, Syrian Private University (SPU), Daraa International Highway, Damascus 36822, Syria

**Keywords:** nanotechnology, *Candida auris*, MDR, antifungal resistance

## Abstract

An emerging multidrug-resistant pathogenic yeast called *Candida auris* has a high potential to spread quickly among hospitalized patients and immunodeficient patients causing nosocomial outbreaks. It has the potential to cause pandemic outbreaks in about 45 nations with high mortality rates. Additionally, the fungus has become resistant to decontamination techniques and can survive for weeks in a hospital environment. Nanoparticles might be a good substitute to treat illnesses brought on by this newly discovered pathogen. Nanoparticles have become a trend and hot topic in recent years to combat this fatal fungus. This review gives a general insight into the epidemiology of *C. auris* and infection. It discusses the current conventional therapy and mechanism of resistance development. Furthermore, it focuses on nanoparticles, their different types, and up-to-date trials to evaluate the promising efficacy of nanoparticles with respect to *C. auris*.

## 1. Introduction

Fungi are eukaryotic organisms that can be found anywhere. They can be found indoors on surfaces and in the air, on people’s skin, inside the body, and outdoors, for example, in soil and on plants. Although there are countless varieties of fungi, only a few of them can truly cause danger to people. Fungal infections are a significant threat to public health as they are associated with life-threatening mycoses and mortality [1,2]. Fungal infections are one of the most common causes of death globally, affecting more than 300 million individuals and resulting in over 2 million deaths annually. In addition, the challenge of mycoses is exacerbated when new pathogenic fungi appear due to their capacity to withstand the few available antifungal drugs, significantly decreasing the efficacy of treatments [3,4,5,6,7]. As seen from this angle, *Candida auris* (*C. auris*) infections have grown to pose a serious hazard to human health worldwide as it is difficult to be diagnosed by conventional laboratory techniques, and some strains are resistant to all kinds of antifungal drugs that are frequently utilized to combat *Candida* infections [8,9]. 

Some fungi are commensal organisms that reside on the skin and in the digestive tract; if they leave their normal environment, they are thought to pose a risk of developing various fungal infections. For instance, the danger of infection spreading has increased due to a dramatic rise in the usage of antibiotics, chemotherapeutic treatments, and immunosuppressive medications [10,11]. Due to the increase in the use of invasive medical devices and procedures (such as catheters and hematopoietic transplantation), these commensal fungi also have a greater potential to enter tissues and blood and cause invasive diseases [12,13]. Additionally, recent substantial health problems unrelated to mycoses, such as seasonal influenza outbreaks and the SARS-CoV-2 pandemic, have exacerbated the intensity of the population’s diseases and susceptibility to secondary fungal infections [14,15,16]. 

When a patient fails to respond or no longer responds to a treatment when it is administered at the advised dosage, therapeutic failure and the development of resistance occur. Several factors lead to therapeutic failure, some related to the patient and others related to the drug. For instance, poor compliance, co-infection, cavitary lesions and abscesses near the site of infection, and obesity may relate to the patient [17]. In addition, immunocompromised patients receiving immunosuppressive drugs are more vulnerable to treatment failure, because the drug is not accompanied by a robust immune response in the fight against infection [18]. Other factors may relate to drugs, such as non-linear pharmacokinetics, drug–drug interactions, selectivity, toxicity, and spectrum of activity [19]. 

There are only five classes of drugs available for treating fungal infections. Since fungi are eukaryotic cells, such as mammalian cells, it is difficult to identify specific therapeutic targets against them. These previous issues drive modern research to nanoparticles, as carriers or adjuvants, to improve the efficacy and performance of current medication [20,21]. This review will provide insights into current trends in nanoparticles and their mechanisms to combat multidrug resistance (MDR) fungal infection.

## 2. Fungal Infections 

Fungal infection can be categorized according to the affected site of infection, into superficial, cutaneous, subcutaneous, mucosal, and systemic infection. The main three pathogenic fungi in humans are *Candida, Aspergillus*, and *Cryptococcus* which account for 90% of fatalities in either immunocompetent or immunodeficient people. *Pneumocystis, Coccidioides*, and *Histoplasma* are three other pathogenic fungi that can seriously harm tissues and even kill people [22]. The species of the infected fungus and the state of the host’s immune system have a significant impact on the type of infection [23]. For example, nearly one billion people have superficial fungal infections, which are among the most prevalent fungal illnesses [24]. Conversely, invasive fungal infections are the most dangerous. These are brought on by inhaling or injecting fungal spores, or by an imbalance of the host’s commensal fungi [25,26].

*Candida* species (spp.) are commensal fungi found on the human skin, mucosa, or intestinal tract; their growth and proliferation are highly restricted in people with a healthy immune system. According to previous studies, the most common pathogenic *Candida* spp. that cause human infections are *Candida parapsilosis*, *Candida albicans*, *Candida krusei*, *Candida glabrata*, and *Candida tropicalis*. Recent research demonstrates that *C. auris* has spread around the world as an MDR fungal infection that significantly increases patient death [27,28]. Furthermore, the CDC’s data show that *C. auris* most closely mimics infectious, MDR bacteria, such as methicillin-resistant *Staphylococcus aureus* (MRSA) [9]. 

*Cryptococcus* spp. cause cryptococcosis which is a widespread invasive fungal infection that poses serious therapeutic difficulties and high fatality rates [29,30]. However, due to advances in molecular science and studies on epidemics, *C. gattii* was recognized as a separate species in 2002 [31,32,33]. Cryptococcal infection can result in pneumonia in immunodeficient patients and is brought on by the inhalation of cryptococcal spores into the lungs. However, in immunocompetent hosts, the infection may be latent without any symptoms. Unfortunately, this cryptococcal infection may spread to any organs, including the brain, and cause lethal cryptococcal meningitis [34,35].

*Aspergillus* is a saprophytic fungus that grows in soil and has over 200 species. *Aspergillus* spp. is widespread and frequently isolated from cultures of the respiratory tracts of asymptomatic individuals. Conversely, invasive aspergillosis leads to chronic obstructive pulmonary disease (COPD) [36,37]. The most frequent species associated with invasive infection, especially in an immunodeficient patient, is *A. fumigatus A. flavus*, *A. niger*, and *A. terreus* also cause invasive infections [38].

### Candida auris Infection

The *Candida* species, a diploid fungus, is regarded as an opportunistic pathogen that can harm people’s health and cause fatal illnesses. Candidiasis is ranked as the fourth nosocomial infection with a high mortality rate ranging from 35 up to 100% in immunodeficient patients [8,39]. *C. auris* is a newly discovered pathogenic fungus that was first discovered in Japan in 2009 [40]. Infections with *C. auris* have been documented in 45 nations in Far East Asia, the Middle East, Africa, Europe, North America and South America, demonstrating its worrying rapid evolution worldwide [9]. Moreover, many clinical reports demonstrated that *C. auris* outbreaks are linked to the COVID-19 pandemic [41,42,43,44,45,46]. 

*C. auris* has been divided into four major separate genetic clades, the South Asian, East Asian, South African, and South American clades, based on geographic origin and genomic data gained by whole genome sequencing and the first isolated locations [28]. Recently, it was revealed that a fifth clade came from Iran [47]. Although *C. auris* is most frequently found on human skin, multiple investigations have shown that the organism may also be isolated from the mucosae of the mouth, esophagus, and gut [28]. Horton et al. reported that *C. auris* produced biofilms on pig skin and in synthetic sweat media that mimicked the physiological circumstances of the axilla [48]. Acquiring infection from contact with soiled surfaces is significantly more troubling, where *C. auris* biofilms have been proven to withstand artificial dehydration [48]. Similarly to other significant *Candida* infections, *C. auris* primarily affects a wide range of vulnerable people, including those with a deficient immune system, a chronic illness such as uncontrolled diabetes, or those taking immunosuppressive medications [49]. Currently, it is documented that *C. auris* can infect people and cause a wide range of illnesses, including fungemia, wound infections, urinary tract infections, meningitis, myocarditis, skin abscesses, and bone infections [50,51]. 

According to the most recent systematic review and meta-analysis study by Chen et al. [52] over 4733 cases of *C. auris* were documented in around 33 countries, with most cases in South Africa, the USA, India, Spain, the UK, South Korea, Colombia, and Pakistan. The majority of cases were identified between 2013 and 2019, peaking in 2016 and then declining after that. Clades I and III were the most common, with more cases documented and an expanded geographical range. Furthermore, 32% of the patients had bloodstream infections which differed based on the clades. The fluconazole, amphotericin B, caspofungin, anidulafungin, and micafungin resistances in *C. auris* were 91, 12, 12, 1.1, and 1%, respectively. The total mortality rate of *C. auris* infections was 39%. Moreover, subgroup analysis revealed that the mortality rate was lower in Europe (20%) and greater in those with bloodstream infections (45%) [52].

*C. auris* shares virulence features with the majority of other *Candida spp.*, including *C. albicans*, *C. tropicalis*, and *C. parapsilosis*, which belong to the CTG clade, or species that translate the CTG codon into serine rather than leucine. [53]. These traits include biofilm formation, yeast-to-hyphae transition, and phenotypic switching [54,55,56]. 

## 3. Current Conventional Medications

There are roughly five classes of conventional antifungal drugs that can be used for topical and systemic antifungal therapies, including azoles, polyenes, echinocandins, allylamines and pyrimidine analogs (Figure 1) [57,58]. Polyenes have been identified as being produced by Streptomyces spp., in which they play a role as a natural defense mechanism. This class includes amphotericin B and nystatin. They work by attaching to the ergosterol present in the fungal cell membrane, generating holes there, and increasing ion permeability. This alters the ion gradient inside and outside the cell membrane, loss of cell integrity, and ultimately results in fungal cell death [59,60]. The most effective polyene for invasive fungal infections is amphotericin B, which works by generating an extra-membranous fungicidal sterol sponge that impairs membrane integrity [61]. Another way by which amphotericin B acts is by the accumulation of reactive oxygen species (ROS), which in turn disrupt the mitochondria, proteins, DNA, and membranes [62,63].

The azole class includes triazole and imidazole [64]. This class acts by blocking the ergosterol synthesis pathway. They bind to and inhibit the lanosterol 14-α-demethylase enzyme that is responsible for the rate-limiting step in the conversion of lanosterol to ergosterol [65,66]. Lanosterol 14-α-demethylase is produced by the genes *ERG11* in yeast and Cyp51 in mold [67].

The echinocandin class mostly includes caspofungin, micafungin, and anidulafungin. They target the β-1,3 glucan synthase and interfere with the integrity of the fungal cell wall [68,69]. The *FKS* family of genes encodes the 1,3-d glucan synthase enzyme [69]. Although the safety profiles of these antifungal drugs are good, the lipid side chains limit their oral absorption. They are effective against both planktonic cells and biofilm-forming cells (sessile cells). Similarly, aspergillosis has been treated with this class [70]. 

Allylamines work by blocking the squalene epoxidase that converts squalene into lanosterol, thus inhibiting the formation of ergosterol and thereby inhibiting fungal growth. They have a broad spectrum of activity and low toxicity [71,72]. 

The antimetabolite, 5-flucytosine (5-FC), is the fifth antifungal class. They enter the fungal cell via cytosine permeases, where it is deaminated to 5-fluorouracil. This prevents the synthesis of both nucleic acids (DNA and RNA), which therefore prevents the synthesis of proteins [73]. Moreover, 5-FC can penetrate the blood–brain barrier to treat fungal infections of the central nervous system [74].

The main challenge for current conventional drugs is to combat and overcome MDR fungi such as *C. auris*, as it is naturally resistant to one or more kinds of commercially available antifungals. Fluconazole is extremely resistant to most *C. auris* isolates, but the minimum inhibitory concentration (MIC) analysis also revealed that certain strains are also resistant to all kinds of antifungal medications [27]. The best method of combating *C. auris* is not yet established. Echinocandins are recommended as the first line of treatment since they are effective against most isolates in the US [75,76]. Additionally, isavuconazole was discovered to be effective against a range of *C. auris* isolates despite their resistance to azoles [75].

## 4. Resistance of *C. auris* to Conventional Antifungals

*C. auris* is a recent trend, unlike other *candida* spp., due to its persistent resistance has subsequently evolved to be MDR-resistant. Furthermore, *C. auris* generates chronic and fatal infections with poor prognoses, especially in susceptible people [77]. According to a study by Osei Sekyere, nearly half of all *C. auris* isolates from various studies exhibited resistance to fluconazole (44.29%), the most commonly used azole antifungal, followed by amphotericin B (15.46%), voriconazole (12.67%), caspofungin (3.48%), and flucytosine (1.95%). Fortunately, it seems that the yeast still responds to echinocandin, so this can be used as the first line of treatment [78]. 

*C. auris* uses a variety of different molecular strategies to bypass the effects of antifungals (Figure 2). Briefly, *C. auris* develops azole resistance by overexpressing or developing a point mutation of the ERG11 gene, which encodes the lanosterol-14-α-demethylase enzyme, preventing azoles from binding their target. Additionally, *C. auris* can reduce the internal concentration of antifungals by overexpressing the MDR-1 gene, which encodes the major facilitator superfamily (MFS) drug exporter pump, and the CDR-1 gene, which encodes the ATP-binding cassette (ABC) drug exporter pump [79,80,81]. 

Echinocandin resistance is developed through mutation or substitution in the *FKS-1* gene that encodes β-1,3 glucan synthase enzyme, which is a critical component in the fungal cell wall [80,82,83]. Furthermore, resistance to amphotericin B is developed through mutation in *ERG11* gene, thereby impairing ergosterol biosynthesis [84]. Amino acid substitution in the *FUR-1* gene (F211I), which is involved in 5-FU metabolism, leads to the development of 5-FC resistance [83].

*C. auris* also can build biofilm and develop resistance to almost all antifungal classes. For instance, in a previous study, Sherry et al. demonstrated that sessile *C. auris* cells have higher MICs for several antifungals than planktonic cells [55]. Unfortunately, the biofilm not only increases the resistance and virulence of inside fungal communities but also enhances the upregulation of the ABC and MSF exporter pumps by 2 to 4 folds [55,81].

It is essential to find new strategies for fighting *C. auris* because of the widespread antifungal resistance and high rates of morbidity and mortality. Nanoparticles appear to offer a promising replacement for resistant drugs. In addition, nanoparticles can be used with antifungal medications to create a powerful synergistic impact that can effectively combat MDR *C. auris*. 

## 5. Nanoparticles (NPs) and Nanotechnology (NT) to Combat MDR *C. auris*


To effectively treat fungal infections and circumvent the fungal multi-resistance to existing medications, the creation of drug delivery systems based on nanoparticles (NPs) is a potential substitute for creating novel pharmaceutical formulations [85,86]. NPs can be constructed from lipids, polymers, or metals [87,88,89]. They offer many advantages over conventional drugs [90,91]. They are more targeted to the site of infection, possess a larger surface area, possess fewer toxic effects and side effects, and rarely develop resistance [92,93,94,95,96]. NPs can be divided into three categories: organic, inorganic, or polymeric. Carbon nanoparticles are further categorized based on their size, shape, chemical composition, and physical characteristics [97,98,99]. Organic NPs are biodegradable, non-toxic, and sensitive to heat and light. Examples include polymers [100,101,102,103], liposomes [104,105], micelles [106]. and dendrimers [107,108]. This type of NP is the first choice in the biomedical field, especially for medication delivery [97,98]. Inorganic NPs can be constructed from metal or metal oxide [109]. Metal NPs commonly include Ag [87,110,111,112], Au [113], Cu [114,115,116,117], Si [118], and Se [119,120]. However, metal oxide NPs are produced when the characteristics of the metal particles are altered in the presence of oxygen, boosting their reactivity and effectiveness. Metal oxide NPs commonly include NO [121], ZnO [122,123,124], CuO [125], TiO_2_ [126], and Fe_2_O_3_ [127]. Moreover, carbon NPs may include black carbon, carbon nanotubes, carbon nanofibers, and graphene [97,98]. 

### 5.1. Metallic NPs

#### 5.1.1. Silver Nanoparticles (AgNPs)

AgNPs are now recognized to have a strong anti-*C. albicans* biofilm action. Previous studies have proven that silver nanoparticles are effective against MDR pathogens and nosocomial infections [128,129,130,131]. Roberto et al. demonstrated that AgNPs can exert promising antifungal activity against MDR *C. auris*, whether present in planktonic form or sessile in biofilm [87]. In their study, they tested different strains from different clades and proved that AgNPs exhibited strong action against the fully developed and preformed biofilm of *C. auris*, regardless of their clade. Additionally, they found that AgNPs have a powerful effect on preventing the production of biofilm by the various *C. auris* strains. Moreover, AgNPs may affect the structure of biofilm in some strains. 

In a parallel study, Lara et al. proved the inhibitory effect of AgNPs against the ability of *C. auris* to develop biofilm on medical surfaces such as silicon elastomer catheters and elastic bandage fibers. They synthesized pure and round AgNPs with a size range of 1 to 3 nm and discovered their dose-related activity against *C. auris* with an altered and disrupted cell wall. Additionally, they showed that elastic bandage wraps maintained the fungicidal action of AgNPs even after numerous washings, demonstrating their long-lasting antifungal potency and efficiency [129]. 

Sheeanana et al. developed a new coating surface system, consisting of a copper sheet coated with a cluster of AgNPs, through an ion exchange reaction and a reduction reaction [132]. This developed surface passed 1 to 7 days of tests for pathogenic *C. auris*. Following the prolonged exposure intervals, it was discovered that more than 90% of the *C. auris* were no longer viable.

In the most recent study, Reem et al. proved the promising activity of AgNPs to combat *C. auris* growth and biofilm formation [133]. In their study, they test the susceptibility of eight isolates of *C. auris* against AgNPs and showed that over 80% of biofilm development was prevented at a comparatively high AgNPs concentration (6.25 g/mL). In contrast, Malik et al. synthesized chemically-stable AgNPs (CC-AgNPs) with a green synthesis method using *Cynara cardunculus* extract as a reducing and capping agent. They tested the potency of their AgNP system against *C. auris* MRL6057 and found that CC-AgNPs can combat *C. auris* through direct inhibition of the cell cycle and arrest the cells in the G2/M phase [134].

#### 5.1.2. Bismuth Nanoparticles (BiNPs)

Vazquez-MunozIn et al. (at 2020) recognized the antibacterial characteristics of elemental BiNPs, especially their anti-candidal activity, particularly against *C. albicans* [135]. In the same year, they derived another study and proved that BiNPs also have potent activity against different strains of *C. auris* [136]. In a later study, they found a significant anti-*C. auris* activity of BiNPs with a MIC ranging from (1 to 4 µg/mL), regardless of their clades. However, BiNPs seemed to have a moderate inhibitory effect on biofilm. Despite this lower activity, BiNPs can alter the biofilm structure and, in some cases, the cell morphology of the cells within biofilms.

#### 5.1.3. Trimetallic NPs

Majid Kamli et al. developed a novel trimetallic NP system (Ag-Cu-Co), using Salvia officinalis leaves [137]. According to their investigation, *C. auris* cells exposed to these trimetallic NPs experienced cell cycle arrest in the G2/M phase, a breakdown of the mitochondrial membrane, the release of an apoptotic marker, and apoptosis at an MIC ranging from 0.39 to 0.78 µg/mL. In addition, compared to their monometallic competitors, Ag-Cu-Co trimetallic NPs have stronger antibacterial characteristics. This is because of the synergistic impact of the Ag, Cu, and Co present in the as-synthesized nanoparticles.

### 5.2. Metal Oxide NPs

Levi Cleare et al. created a novel N-acetylcysteine S-nitrosothiol NP (NAC-SNO-NP) system that promotes a prolonged release of nitric oxide (NO) [121]. Using this NP model, they want to mimic the natural NO which is considered an important component in the innate immune system and possesses cytotoxic activity against a variety of pathogens [138,139,140]. They demonstrated that this NP system can perfectly reduce the growth of *C. auris* and decrease the development of biofilm by more than 70% at 10 mg/mL [121]. Notably, the NP architecture itself exhibited an intrinsic inhibition of *C. auris*, demonstrating that the antifungal activity was a combined consequence of the NP itself and the released NO. In another study, Vargas-Cruz et al. prepared a nitroglycerin–citrate–ethanol (NiCE) catheter lock solution and evaluated its efficiency in eradicating *C. auris* biofilms in central line lumens by converting nitroglycerin into NO [141]. Additionally, they compared the effect of the NiCE catheter lock solution with widely accepted antifungal drugs, such as caspofungin, micafungin, voriconazole, liposomal amphotericin B, and others, and proved that NiCE possesses a superior effect in eradicating *C. auris* biofilms [141]. 

Moreover, Sherin Philip et al. synthesized iron oxide (Fe_2_O_3_) NPs that were stabilized by supramolecular β-cyclodextrin and evaluated their activity in combatting *C. auris*. They showed that Fe_2_O_3_ NPs can inhibit *C. auris* with an MIC of around 500 µg/mL [142].

### 5.3. Nanofibrous Membrane

Liu et al. generated a novel form of polylactic acid-hypocretin A (PLA-HA) nanofibrous membrane. They conducted in vitro and in vivo studies to evaluate the PLA-HA-based antimicrobial photodynamic therapy (aPDT) effects in combatting *C. auris* infection [143]. aPDT is a novel antimicrobial strategy that uses a non-toxic photosensitizer (PS) and appropriate light sources to stimulate the generation of reactive oxygen species (ROS), which can destroy pathogenic microbes [144,145]. Hypocrellin A (HA) is a natural lipid-soluble pigment that belongs to the perylenequinonoid class and is considered a novel form of PS [146]. Liu and his colleagues provided evidence that PLA-HA is an effective antifungal agent for treating superficial *C. auris* infections. They concluded that this is because intracellular ROS generation causes yeast cells to die [143].

### 5.4. NPs Loaded with Commercially Available Antifungal Drugs

In a recent novel study, Henry et al. synthesized chitosan-(poly lactide co-glycolide) NPs (C-PLGA NPs) as a nanocarrier system and loaded it with fluconazole. The sustained drug release from this nanocarrier system is pH-dependent. For instance, at a pH of 7.0, 34% of the release happened and at a pH of 4, 83% of the release happened [147]. Moreover, they evaluate the efficacy of C-PLGA-loaded NPs versus MDR *C. auris* and demonstrated that this nano-formulation significantly increases the antifungal activity up to 64-fold compared to conventional fluconazole [147]. Fayed et al. synthesized zinc oxide NPs loaded with caspofungin and demonstrated that these loaded NPs can prevent the phenotypic changes in *C. auris* that lead to the development of caspofungin resistance [148]. 

In another study, Gabriel Davi et al. developed a nano-emulsion system and loaded it with amphotericin B. Additionally, they tested its antifungal potency against *C. auris* using an in vivo model of *Galleria mellonella* and proved the significant activity of this loaded nano-emulsion system compared to free amphotericin B [149]. 

In a similar manner, the same team conducted another trial using micafungin-loaded nano-emulsion and test its in vitro and in vivo efficacy and toxicity using *Galleria mellonella* model [150]. 

### 5.5. NPs Loaded with Natural Drugs

Essential oils (EOs) are an effective option for treating fungi and acting as a modulator of fungal biofilms [151,152]. De Alteriis et al. encapsulated *Lavandula angustifolia* EOs, extracted from a lavender plant, in liposomes and investigated their effectiveness against *C. auris* persister-derived biofilm. They concluded that this loaded liposome could combat both primary and persister *C. auris* biofilm through the production of ROS that may affect the expression of certain genes involved in biofilms [153]. 

### 5.6. Nanotechnology (NT) for Diagnosis of C. auris

Luis et al. constructed a novel system for the selective and sensitive detection of *C. auris* in clinical samples using a nanoporous anodic alumina (NAA) biosensor that had been encapsulated with oligonucleotides. The NAA support is firstly packed with rhodamine B, a fluorescent reporter dye, and then capped with a variety of oligonucleotide sequences that precisely hybridize with distinct regions of the *C. auris* genome. Therefore, the capping oligonucleotide prevents dye release by obstructing pores. In the presence of *C. auris* genomic DNA, the capping oligonucleotide is displaced (due to favorable oligonucleotide–DNA hybridization), uncapping the pores and permitting dye transportation. Through this system, *C. auris* can be detected at concentrations as low as 6 CFU/mL, making it possible to diagnose clinical samples in just one hour without the need for DNA extraction or amplification procedures first [154].

## 6. Expected Mechanisms of NPs to Combat MDR *C. auris*

The exact mechanism of action of NPs is not known; however, many reports and studies suggest the general mechanisms of free NPs or loaded NPs to exert their antimicrobial activity. In general, the size, shape, and coating agents of NPs have a significant impact on their antifungal activity. Firstly, NPs interact with the outer surface of fungi and form aggregates, leading to the formation of pits in the cell wall. As a result, a decrease in membrane permeability and loss of membrane fluidity may occur, resulting in a disruption of energy transmission and cell death (Figure 3). Formed pits let NPs enter the fungal cell. Once entered, they lead to the accumulation of ROS that trigger and enhance apoptosis. ROS can disrupt macromolecules in the cell, resulting in lipid peroxidation, protein modification, enzyme inhibition, inhibition of the electron transport chain, RNA or DNA damage, and therefore cell death [155,156,157].

NPs may bind to and disrupt vital cell components. Furthermore, they may interrupt significant intracellular signaling pathways [137,156]. They can penetrate the biofilm structure and may change the cell morphology or disrupt sessile organisms within biofilm [87].

Specifically to AgNPs and splatted Ag^+^, the essential functions of fungal cells are considerably changed by the modulation of the transcriptome, epigenome, and metabolome. Moreover, they may cause the down-regulation of the genes involved in the tricarboxylic acid cycle, redox metabolism, ergosterol production, and lipid metabolism causing structural alterations, primarily at the level of biological membranes [158,159,160].

Synergistic activity may occur when NPs are loaded with antifungal drugs. In this case, they act by dual mechanisms: they transport antifungal drugs to the target site, provide a large surface area of both NPs and antifungal drugs, leading to more toxic action on fungal cells, and combat MDR pathogens [147,161] (Figure 3).

## 7. Conclusions and Future Perspectives

*C. auris* is a newly emerged fungus and may cause outbreaks of nosocomial infection. *C. auris* infections have become a severe threat to human health across the world because they are difficult to identify using normal laboratory approaches and certain strains are resistant to all antifungal classes. Thus, alternative therapies that are both safer and more effective are urgently needed. Moreover, the increase in MDR fungal infections and the scarcity of clinically effective antifungal drugs signals the need for the development of new antifungal approaches to manage these issues in the context of a future that is already challenging.

NPs seem to be a promising approach to combatting and overcoming MDR fungi, such as *C. auris*. Although recent studies demonstrated the promising effect of NPs on combatting *C. auris* infection, the applications of NPs will not be ready until future studies emphasize their pharmacokinetic and pharmacodynamic profiles, physicochemical interactions, toxicities, and specific mechanisms of action.

## Figures and Tables

**Figure 1 pathogens-12-01033-f001:**
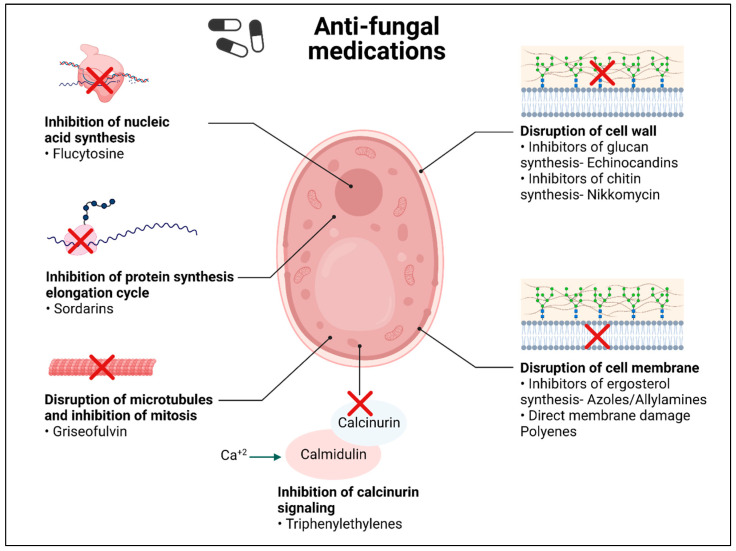
Current conventional anti-fungal drugs and their mechanism of action. (Created with BioRender).

**Figure 2 pathogens-12-01033-f002:**
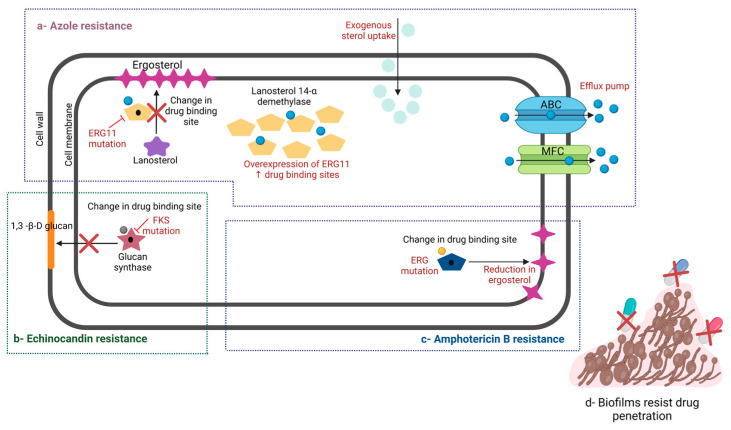
Development of resistance in *C. auris* against conventional medication. (**a**) *C. auris* develops resistance to azoles through overexpression and point mutation in *ERG11* gene, which encodes the lanosterol-14-α-demethylase enzyme. (**b**) *C. auris* develops resistance to echinocandins through mutation or substitution in *FKS-1* gene, which encodes the β-1,3 glucan synthase enzyme. (**c**) *C. auris* develops resistance to polyenes through mutation in ERG and so impairs ergosterol biosynthesis. (**d**) *C. auris* also can build biofilm and develop resistance to almost all antifungal classes. (Created with BioRender).

**Figure 3 pathogens-12-01033-f003:**
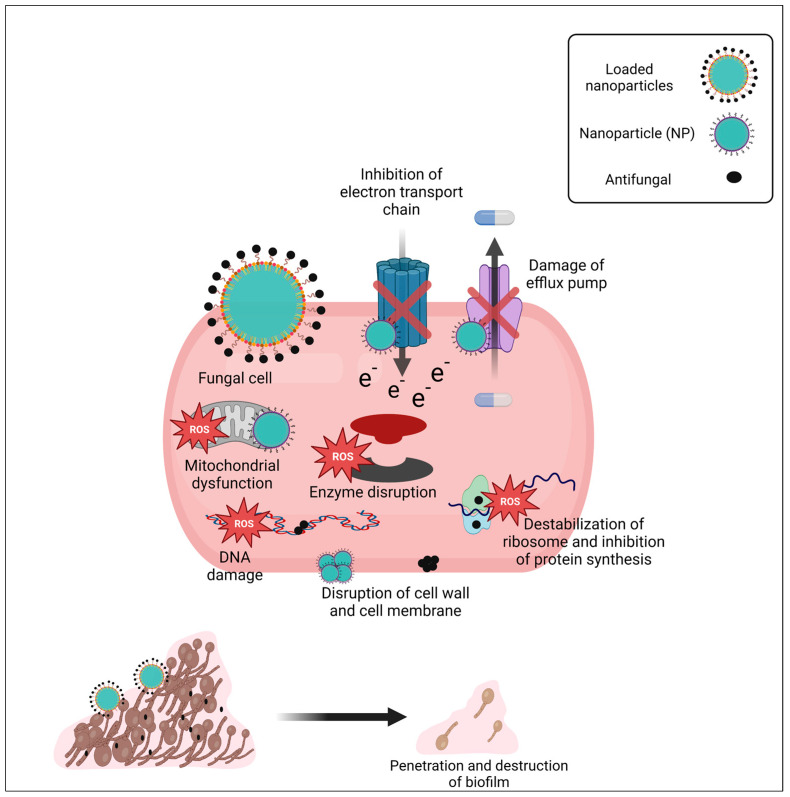
Expected mechanism of free and loaded NPs for combatting *C. auris* infection. In the case of free NPs, they accumulate at the outer surface of the cell, disrupt the cell wall, and form pits through which they enter the cell and then disrupt the cell membrane, resulting in a decrease in membrane permeability, and a loss of membrane fluidity may occur, resulting in a disruption of energy transmission and cell death. Once NPs enter through formed pits, they bind to and disrupt vital cell components and interrupt significant intracellular signaling pathways. On the other hand, NPs increase oxidative stress, which leads to an accumulation of ROS, which have the capacity to disrupt macromolecules in the cell, resulting in lipid peroxidation, protein modification, enzyme inhibition, inhibition of electron transport chain, and RNA or DNA damage, thereby promoting cell death. In case of loaded NPs, they carry antifungal drugs and facilitate their transport to its specific targets inside the fungal cell. Hence, loaded NPs possess a synergistic activity, transport antifungal drugs to the target site, provide a large surface area of both NPs and antifungal drugs, leading to more toxic action on fungal cells, and combat MDR pathogens, such as *C. auris*. Moreover, *C. auris* tends to form biofilm and become resistant to conventional antifungals. Loaded NPs have the capacity to the penetrate extracellular matrix of biofilm, transport antifungal agents inside the biofilm, and exert their fungicidal effect. (Created with BioRender).

## Data Availability

Not applicable.

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
