# Peer review of "Nanotechnology-Based Strategies to Combat Multidrug-Resistant Candida auris Infections"

_pathogens, 2023, doi:10.3390/pathogens12081033_

Round 1

Reviewer 1 Report

Page 3 line 95,- please put the meaning of MRSA

Page 3 line 100.-  cryptococcal infection should be Crytococcal infection

Page 3 line 109.- A. fumigatus. A. flavus should be A. fumigatus, A. flavus

Please consider to put 2.1 Candida auris infection instead  3. Candida auris infection. I think it is the subject of section 2.  Or, failing that, redraft the section about candida infections including Candida auris from point 2.

In figure 2 the letters a,b,c,d do not appear, it would be convenient to put them.

Figure 3 Please indicate with a box A and B the effect of nanoparticles on cells and biofilm. Point to the nanoparticle. The figure is not clear with respect to the figure caption. Which is the free nanoparticle and which is the charged one?

I think that point 7 looks like a summary of the points described in section 6. I think that it could be eliminated by improving figure 3. This figure could be a good illustration of point 6

Some references lack doi please check.

Author Response

Reviewer 1:

First, we would like to thank the reviewer for the efforts and time spent in careful revise of our manuscript. We believe that these comments will enhance the caliber of our work.

Comments and Suggestions for Authors

Page 3 line 95,- please put the meaning of MRSA

Response: done.

Page 3 line 100.-  cryptococcal infection should be Crytococcal infection

Response: done.

Page 3 line 109.- A. fumigatusA. flavus should be A. fumigatusA. flavus

Response: done.

Please consider to put 2.1 Candida auris infection instead  3. Candida auris infection. I think it is the subject of section 2.  Or, failing that, redraft the section about candida infections including Candida auris from point 2.

Response: done

In figure 2 the letters a,b,c,d do not appear, it would be convenient to put them.

Response: done.

Figure 3 Please indicate with a box A and B the effect of nanoparticles on cells and biofilm. Point to the nanoparticle. The figure is not clear with respect to the figure caption. Which is the free nanoparticle and which is the charged one?

Response: done

I think that point 7 looks like a summary of the points described in section 6. I think that it could be eliminated by improving figure 3. This figure could be a good illustration of point 6

Response: we tried our best to improve it. We believe that this section is important to highlight the mechanisms of NPs to combat MDR C. auris.

Some references lack doi please check.

Response: done.

please see the attachment file 

Reviewer 2 Report

The manuscript authored by Helal F. Hetta et al., entitled Nanotechnology based Strategies to Combat Multidrug Resistant Candida auris Infection encompasses sound and brief information.  However I would like to make few suggestions for the improvement of the proposed review which are as follows:

1. The introduction lacks bibliographic information of last 15 years of studies published in the aforementioned domain. Try to use graphical charts or pie charts with significant justification and explanation for the same to support your study.

2. The manuscript fails to mention the incidence and importance of the study with respect to different geographical boundaries. 

3. Discuss cohort study cases if performed previosly.

4. Improve the figures using three dimensional objects and give source citation of the tools and softwares used.

5. Keep the manuscript brief and crisp and improve the grammatical errors wherever required.

6. Make short and brief sentences with logical information.

7. Improve Future and Conclusion section with literature support.

8. Draw a schematic of mechanistic inhibition of nanoparticle mediated mode of action against C. auris.

Moderate changes are required.

Round 2

Reviewer 1 Report

The article has been improved. In this paper version figure 2 is missing, please put it.

Reviewer 2 Report

The authors should keep in mind while preparing the rebuttal letter they should try to mention the line numbers within the manuscript where suggested changes have been incorporated.